# Collecting Verified COVID-19 Question Answer Pairs

**Adam Poliak[1], Max Fleming, Cash Costello, Kenton Murray, Shivani Pandya,**
**Darius Irani, Milind Agarwal, Udit Sharma, Shuo Sun, Nicola Ivanov,**
**Mahsa Yarmohammadi, Lingxi Shang, Kaushik Srinivasan**
**Seolhwa Lee, Xu Han, Smisha Agarwal, João Sedoc[2]**
[1]Barnard College, Johns Hopkins University, [2]NYU Stern School of Business
`https://covid-19-infobot.org/`

## Abstract

We release a dataset of over 2,100 COVID-19 related Frequently asked Question-Answer pairs scraped from over 40 trusted websites. We include an additional $24,000$ questions pulled from online sources that have been aligned by experts with existing answered questions from our dataset. This paper describes our efforts in collecting the dataset and summarizes the resulting data. Our dataset is automatically updated daily and available at `https://github.com/JHU-COVID-QA/scraping-qas`. So far, this data has been used to develop a chatbot providing users information about COVID-19. We encourage others to build analytics and tools upon this dataset as well.

## 1 Introduction

With the quick spread of COVID19, misinformation has rapidly spread.[1] Misinformation around the use of certain drugs for the prevention of Covid-19 has had fatal outcomes, and stigmatization guided by misinformation about certain communities as vectors of virus undermines the long-term welfare of our society. We are developing a natural language processing (NLP) backed-informational chatbot targeted at comprehensive COVID-19 information and misinformation. Users can interact with our chatbot on different platforms to access information about COVID-19, available care, and other topics of interest.[2]

To aid in this effort, we aggregate factual information in the form of verified questions and answers to help answer frequently asked questions about the pandemic. We employ three main aggregation efforts in tandem: 1) generating high quality and accurate information from domain experts,

i.e. Public Health researchers at Johns Hopkins University; 2) automatically scraping frequently asked questions and answers from online trusted sources, e.g. newspapers and government agencies; and 3) automatically ranking and manually aligning additional questions from social media with the scraped questions and answers in our dataset. This paper primarily describes our efforts to extract high quality content from trustworthy websites and domain experts. Our effort has resulted in a publicly available dataset that currently contains over 2,100 Questions and Answers from more than 40 webpages. The dataset is available at `https://covid-19-infobot.org/data/`. Since we are actively scraping more websites and re-scrape all sites at least once a day, these numbers are updated daily.[3]

## 2 Creating our FAQ Dataset

We create our publicly available dataset of over 2,100 question-answer pairs by aggregating FAQs from trusted news sources.[4] We choose websites to scrape based on three broad criteria: 1) the informativeness and trustworthiness of the website; 2) the ease of scraping frequently asked question-answer pair from the website; and 3) the number of questions and answers on the website.

We use a straightforward scraping process that enables undergraduate students to contribute to our efforts. We developed a python library for students to easily add scrapers to our project. As demonstrated in the example in Figure 1, our library requires each question-answer (and metadata)

---

[1]`https://www.newsguardtech.com/coronavirus-misinformation-tracking-center/`
[2]`https://covid-19-infobot.org/`

[3]The dataset's statistics described in this paper are based on a snapshot of the data as of June 25th, 2020, corresponding with `https://github.com/JHU-COVID-QA/scraping-qas/tree/a446c00c318e02cad5188cec359b9d649d8c4933`

[4]We additionally have over 300 question-answer pairs manually created by Public Health experts. We plan on including these in our publicly available dataset at a later date.

```
converter.addExample({
    'sourceUrl': 'example.com',
    'sourceName': "example",
    "needUpdate": True,
    "typeOfInfo": "QA",
    "isAnnotated": False,
    "responseAuthority": "",
    "question": '<a href="example.com/dir1">What is COVID-19?</a>',
    "answer": '<p><a href="example.com/dir2">Coronaviruses</a> are a large family of viruses.</p>',
    "hasAnswer": True,
    "targetEducationLevel": "NA",
    "topic": ['topic1', 'topic2'],
    "extraData": {'example extra field': 'example value'},
    "targetLocation": "US",
    "language": 'en',
})
```

Figure 1: Screenshot of our documentation describing the data and metadata stored for each scraped question-answer pair.

to be stored as a simple dictionary. The library automatically adds this information to our set of question-answer pairs. Additionally, the library accordingly handles updating answers to questions in our dataset if a previously scraped website updates its information.

This has enabled students to efficiently join the project and contribute immediately. Further documentation is available at `https://github.com/JHU-COVID-QA/scraping-qas` and we encourage others to join our efforts.

## 2.1 Metadata

For each scraped question-answer pair, we extract relevant metadata for our chatbot and other NLP analytics. The metadata includes information about the source of each question-answer pair (we include both the source name and the URL) and the date when the question-answer was last scraped from or updated on the website. Additionally, if the information on the website is targeted for a specific geographic area, we include that in our metadata as well.

## 2.2 Leveraging existing scrapers

We leverage existing scrapers for collecting questions-answer pairs for COVID-19. 874 of our examples come from scrapers released by deepset.[5] Following deepset's lead, we open-source our scrapers as well.

## 2.3 Continuous scraping

As our understanding of COVID-19 rapidly evolves, trustworthy sources update the informa-

tion they release. Therefore, each day, we automatically re-run the web scrapers to find new information. This enables us to add new question-answers or update answers to existing questions in our dataset.

If a previously scraped question-answer is removed from a website, we remove that example from our dataset.[6] Question and answers that we removed from our dataset as still available in our history since we archive each day's dataset. In turn, the quality of our dataset is constantly evolving and improving.

## 3 Data

The described effort resulted in a dataset that is evolving daily. The June 15th version contains over 2,100 questions and answers scraped from 40 websites. We list the number of question-answer pairs extracted from each source in Table 1. Our dataset contains some examples in different languages besides for English, owing to deepset scraping websites in multiple languages. Figure 2 plots the number of question-answer pairs in each of the five languages: English, German, Polish, Italian, and Swedish. Roughly 70% of our examples are in English. As we release more data, we will include further analysis of the growing dataset.

Websites might update or change how they store information. This is why the current version of our dataset contains just 1 example from the Delaware State Government webpage. The May 20th version of our dataset contains 22 examples from this website.

---

[5]https://github.com/deepset-ai/COVID-QA/

[6]We assume that a website will remove information about COVID-19 that is no longer accurate.

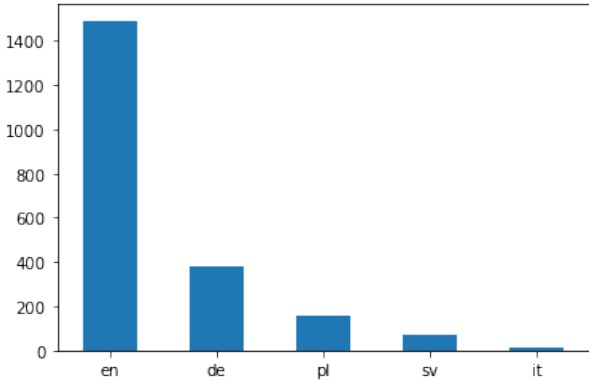

Figure 2: Number of question/answers for each language in our dataset.

## 4 Manually Aligning Additional Questions and Answers

Since the internet contains many more questions that are not answered, we additionally collected questions and align them with the question-answer pairs in our dataset. We leverage information retrieval techniques to match these unanswered questions with questions in our dataset and then rely on domain experts to verify each aligned question-question-answer (QQA) pair. In this section, we provide details for each of these steps.

### 4.1 Online Question Extraction

We downloaded 28 million tweets from the COVID-19 Twitter Dataset (Chen et al., 2020), Qorona,[7] and CovidFaq[8], extracted the questions from those resources,[9] sorted them by frequency, and discarded the questions that occurred less than four times. Then, we grouped semantically similar questions into 9,200 clusters. Next, we extracted the centers of the clusters and, using a state-of-the-art sentence re-writer (Hu et al., 2019), we generated three high quality paraphrases of each question. This resulted in a collection of over 27,000 unanswered questions about COVID-19.

### 4.2 Aligning Extracted Questions with Existing Questions and Answers

We worked with public health experts to align these unanswered questions with our verified question-

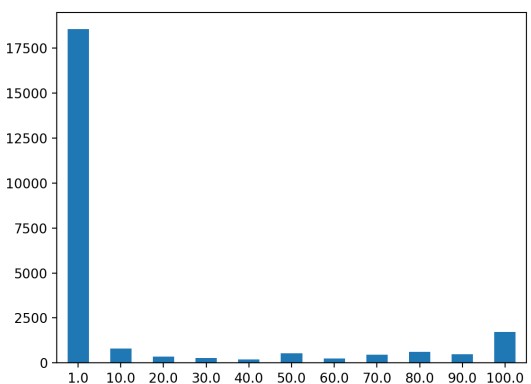

Figure 3: Histogram of number of QQAs (Y-axis) annotated with a score at most the corresponding x-axis. Over 17.5K examples are labeled between 0 and 1.

answer pairs (section 3). For each of these 27,000 questions, we used a BM25 model (Robertson and Walker, 1994; Robertson et al., 1996) to determine the most similar answered questions in our dataset.[10] Following the EASL annotation protocol (Sakaguchi and Van Durme, 2018), for each unanswered twitter question, we presented public health experts with the five most similar QA's from our dataset. Based on a formal protocol developed by a senior Public Health researcher on our team (Figure 4), we asked the experts to determine, on a scale from 0 to 100, how relevant or similar the QA from our dataset is to the unanswered question.

For this annotation effort, we leveraged Turkle, open-sourced, locally hosted clone of Amazon Mechanical Turk developed by the JHU Human Language Technology Center of Excellence.[11] Figure 5 and Figure 6 illustrate our annotation interface.

As part of this protocol, expert annotators could indicate whether a question was not relevant to COVID-19 or whether an existing answer was no longer correct. We removed such labeled examples from our set. This effort results in 24,240 annotated QQAs. Figure 3 plots the distribution of labels annotated for QQAs. Over 18,000 examples were judged to be less than 1% relevant, indicating that the majority of the questions extracted from twitter are irrelevant to the answered questions in

---

[7]https://github.com/allenai/Qorona
[8]https://github.com/dialoguemd/covidfaq

[9]Corona and CovidFaq specifically contain questions. We extract questions from the Twitter dataset by determining whether a sentence from a tweet either ends with a question mark, or starts with a provided list of words (e.g., "who", "when", "where", etc).

[10]We trained the BM25 model by using the answers that previously were manually aligned by experts with the candidate questions. We then calculated scores between the terms in the input question and terms in the candidate answers. We used the implementation in Elasticsearch and relied on the default parameters.

[11]https://github.com/hltcoe/turkle

our dataset. These additional examples can be used to further train a chatbot to answer questions about COVID-19.

## 5 Conclusion

We have presented our growing dataset of over 2,100 question-answers that has been created by scraping over 40 websites. We also discussed other data we collected and annotated that may be beneficial to others in the community as well. Our evolving dataset is complementary to other recent COVID-19 QA datasets, e.g. Tang et al. (2020)'s 124 question-article pairs, Wei et al. (2020) 1,690 questions and 403 answers, and Möller et al. (2020)'s dataset.

## Acknowledgments

We thank the reviewers for their insightful comments. This work was supported in part by DARPA KAIROS (FA8750-19-2-0034). The views and conclusions contained in this work are those of the authors and should not be interpreted as representing official policies or endorsements by DARPA or the U.S. Government.

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

| Source Name | # of Questions-Answers |
|---|---|
| North Dakota Stake Government | 305 |
| Vermont Department of Health | 151 |
| NYTimes | 118 |
| CNN | 106 |
| Kansas Department of Health and Enviroment | 92 |
| FDA | 76 |
| Oregon Public Health Division | 75 |
| Johns Hopkins Bloomberg School of Public Health | 57 |
| FloridaGov | 45 |
| Texas Human Resources | 40 |
| National Foundation for Infectious Diseases | 33 |
| AVMA | 32 |
| WHOMyth | 29 |
| Cleveland Clinic | 29 |
| Public Health Agency of Canada | 28 |
| Ministero della Salute, IT | 16 |
| JHU Medicine | 13 |
| JHU HUB | 7 |
| Hawaii State Government | 4 |
| Delaware State Government | 1 |
| GOV Polska | 154 |
| Bundesministerium für Gesundheit (BMG) | 201 |
| FHM, Folkhälsomyndigheten | 142 |
| Ministero della Salute, IT | 16 |
| World Health Organization (WHO) | 121 |
| Bundesministerium für Wirtschaft und Energie | 34 |
| Berliner Senat | 48 |
| European Centre for Disease Prevention and Control | 47 |
| Bundesanstalt für Arbeitsschutz und Arbeitsmedi... | 35 |
| Bundesministerium für Arbeit und Soziales (BMAS) | 32 |
| Bundesagentur für Arbeit | 11 |
| Presse- und Informationsamt der Bundesregierung | 16 |
| Center for Disease Control and Prevention (CDC) | 13 |
| Robert Koch Institute (RKI) | 4 |
| total | 2115 |

Table 1: Number of question-answer pairs for each source in the dataset scraped on June 25th. Some of these sources contain more than one website. The bottom half represents the websources in our dataset that we extract using deepset's scrapers.

## *Protocol*

There is a lot of subjectivity that enters the picture when we are assigning relevancy through the use of a 0-100% scale. To help standardize our thought process and understanding on how to rank relevancy, we've tried to do it by parsing out systematically what each question is discussing.

Please note that these examples are only showing it one at a time. As you can see in the description of the task and the picture above, it will actually be one user question, and five relevant questions to review

### *Example 1:*
**User  Question:** How many people have it in my town
**Relevant Question**: How long do people have to isolate for?
**Thought Process**: The relevant question is focused on COVID-19, isolation, and duration. The new question is focused on COVID-19 and prevalence.
**Relevancy Scale:** 0% relevant
  ● They are both talking about COVID-19, but don't overlap significantly beyond that.

### *Example 2:*
**Relevant Question:** It would be great to hear more about the symptoms. Cough and difficulty breathing isn't too specific (especially during allergy season!).
**New Question:** Is cough but no fever a symptom?
**Thought Process:** Both of these questions are asking about COVID-19 symptomatology - however the first is discussing how it is different from other illnesses (like the flu, cold, and allergies). The second question is specifically focused on COVID-19 symptoms solely, and what those are.
**Relevancy Scale**: 80-90% relevant
  ● This is because they are both talking about COVID-19 symptoms, but the answers to the questions differ slightly.

Figure 4: Protocol and examples for the expert annotators to align unanswered questions from Twitter with the question-answer pairs in our dataset.

**User Question: does coronavirus like cold or heat**

☐ Invalid/Useless Question

---

**Relevant Question #1: Will warm weather / summer / heat stop outbreak of COVID-19?**

**Relevant Answer:** We do not know. Some viruses, like the common cold or flu, spread more during the cold weather months but people still become sick in

+ Show More

Source   JHU Public Health
Annotator   Shivani Pandya or Smisha Agrawal

Score: 50

Not Relevant ▭ Exactly Relevant

☐ Relevant Question or Answer is **wrong** and needs to be **updated** ?

---

**Relevant Question #2: How long does the virus live on surfaces, like packages, plastic, etc? Should we be worried about that?**

**Relevant Answer:** A recent study shows that the virus can live in the air and on surfaces from several hours to several days. On plastic, it can live for up to 72

+ Show More

Source   JHU Public Health JHU
Annotator   Shivani Pandya or Smisha Agrawal

Score: 50

Not Relevant ▭ Exactly Relevant

☐ Relevant Question or Answer is **wrong** and needs to be **updated** ?

---

**Relevant Question #3: What does it mean to be isolated?**

**Relevant Answer:** This means the person has been instructed to separate themselves from others, to avoid spreading COVID-19 further with others. If you

+ Show More

Source   JHU Public Health CDC
Annotator   Shivani Pandya or Smisha Agrawal

Score: 50

Not Relevant ▭ Exactly Relevant

☐ Relevant Question or Answer is **wrong** and needs to be **updated** ?

Figure 5: Screenshot of our expert annotation interface.

Figure 6: Screenshot of our expert annotation interface.