# OpenReview forum: "Collecting Verified COVID-19 Question Answer Pairs"
_EMNLP/2020/Workshop/NLP-COVID — NLP-COVID19-EMNLP Poster_

### Official Review · AnonReviewer3 · 2020-09-16
**This paper presents a dataset and an high level description of the techniques used to create it. However since the data is not available the significance of this work cannot be assessed.**

**Rating:** 6
**Confidence:** 3

**Review:**

The paper presents a potentially useful dataset of COVID-19-related frequently asked question-answers. The link given in the paper does not yet allow the access to the dataset. (https://covid-19-infobot.org/data/) Presumably this will be made available upon publication?

Introduction: When explaining the three aggregation efforts, what does the 1st mean by "generating high quality information"?
In this sentence “To aid in this effort, we aggregate factual information in the form of verified questions and answers to help answer frequently asked questions about the pandemic.”
Better to use question-answer pairs instead of “questions and answers”.


Section 2:  This sentence is vague: "abstract away adding this information to our set". When updating a question-answer pair how the update is done. The paper lacks the explanation of how the question-answers are updated.

Figure 2 is not a clear example. The answer to the question of "What is COVID-19?" is not correct – COVID-19 is an acronym for “Coronavirus Disease 2019”. The answer is a URL, if the correct answer is to be found there, then perhaps an example of the website content should also be included. The example also does not have any indication of the quality of the answer, and the impression gained is that this might not be a very useful resource at all. It might be more accessible to show a few records in the database, or at least to use a very accurate example from the documentation together with a snapshot of the answer that would be made available. The figure's title indicates the inclusion of metadata, but it does not have the "date/ update date" metadata mentioned under section 2.1.

Section 3: Repeated the “We list the the number of question-answer”.

Section 4: Need correction “We additionally collection”?
Here Qorona and CovidFaq are mentioned, but there is no other reference to them later as if the questions from these sources are used or not? If yes how? If no, why they are mentioned?

Section 4.1 After Twitter data is collected, what technique/tool is used to extract questions? What indicated that a tweet does contain a question?
What techniques is used to group semantically similar questions? Is it topic modelling or something else?
The process of extracting questions from Twitter seems to be the main part of this research but is not explained properly to be useful to the research community.

---

> ### Author Response · Authors · 2020-09-27
> **Response to AnonReviewer3**
>
> Thank you for your detailed and insightful questions! Thank you also for the stylistic issues, e.g. the repeated “the the”. We appreciate your suggestion for using question-answer pairs instead of “questions and answers.” We will make these changes in the updated version of the submission.
>
> When adding a scraped question-answer pair to our dataset from a specific source, we first look to see whether our dataset contains the question from the specific source. If we do not have this question (from this specific source) in our dataset, we simply add the question-answer pair to our dataset. If we have seen this question before, we  replace the answer for the question in our dataset with the newly scraped answer. This process in implemented in this function (https://github.com/JHU-COVID-QA/scraping-qas/blob/e579c2a12e8721fa66842856cdf30dfaac70091f/src/scraping/covid_scraping/utils.py#L93-L152) in our python library.
>
>
> Corona and CovidFaq are included in the list of 27,000 unanswered questions about COVID. We will update the description to include this since “from twitter” should not be included in “This resulted in a collection of over 27, 000 unanswered questions about COVID19 from twitter”. Thank you for catching this.
> Corona and CovidFaq specifically contain questions. We extract questions from the Twitter dataset by determining whether a sentence from a tweet either ends with a question mark, or starts with a provided list of words (e.g., "who", "when", "where", etc.). The selected questions are tokenized. We will release our documentation and code for this process upon publication.
>
> As just mentioned in our response to reviewer #1, although we have not linked the data on our website (you are indeed correct that we were waiting for publication to do this), the data is publicly available on github at https://github.com/JHU-COVID-QA/scraping-qas/. The statistic and description of the dataset included in our submission correspond to this commit (92b68f61671bed63fde1aa927e22a7cccbb821b1) of the dataset. By automatically scraping and committing the updates daily to our dataset, we are enabling those interested to explore how the questions and answers in our dataset change over time.

---

### Official Review · AnonReviewer1 · 2020-09-20
**A COVID-QA dataset is presented but its selection criteria and usage need to be better explained**

**Rating:** 6
**Confidence:** 4

**Review:**

This paper describes the curation efforts to create a QA dataset on COVID-19. It collected 2,200 question-answer pairs from popular websites. Then it extracted 27, 000 unanswered questions from tweets, identified the most similar questions in the collection and provided top 5 answers. Then the provided answers were further manually annotated (whether they are relevant). Please see my comments below.

For the 27, 000 unanswered questions, it seems that only the questions similar to the existing questions in the collection can be kept. If an unanswered question is a new topic of COVID-19, the top retrieved answers will be irrelevant, and health experts will not provide new answers based on the description. Therefore, many new questions will be potentially missed. And what is the motivation of only annotating the questions that are similar to the existing questions in the collection? Critically, the questions that are significantly different from the existing questions are more valuable and need the power of manual curation. Adding these questions will enrich the datasets. In contrast, if a question is similar to the existing questions, adding it will just have one more similar instance.

Second, the study needs to explain how to use this dataset in detail. For instance, the statement 'Over 18, 000 examples were judged to be less than 1% relevant, indicating that the majority of the questions extracted from twitter are irrelevant to the answered questions in our dataset' seems that most of the questions do not have a precise answer yet. How to train a QA model in this case?

Third, from the methodology level, while BM25 is effective, it does not capture the semantics of the questions. Did the study also try to use other methods to calculate the semantic similarity between the questions such as using word or sentence embedding?

Also importantly, the dataset is not public for now. More specific comments cannot be made.

---

> ### Author Response · Authors · 2020-09-27
> **Response to AnonReviewer1**
>
> Thank you for your detailed and thoughtful review.
>
> We have annotations for over 24K question answer pairs. These are publicly available at https://github.com/JHU-COVID-QA/scraping-qas/blob/master/data/annotations/aligned_question_question_answer.csv and will be added to our website upon publication. We will clear up confusion in the updated version of our submission.
>
> There are multiple ways one can train a QA model in the case where over 18K examples were judged to be less than 1% relevant. We trained our QA model by using the cosine similarity between the sentence embedding of an input question and the sentence embedding of a candidate question or candidate answer. Other ways to train a QA model include sampling a smaller subset of the 18K examples, weighting each example based on the distribution of labels, or using a learning to rank setup.
>
> Yes, we have explored other methods for calculating the semantic similarity between the questions. In particular, we have used fine-trained BERT models trained on our dataset.
>
> Although we have not linked the data on our website, the data is publicly available on github at https://github.com/JHU-COVID-QA/scraping-qas/. The statistic and description of the dataset included in our submission correspond to this commit (92b68f61671bed63fde1aa927e22a7cccbb821b1) of the dataset. By automatically scraping and committing the updates daily to our dataset, we are enabling those interested to explore how the questions and answers in our dataset change over time.

---

### Official Review · AnonReviewer2 · 2020-09-25
**Interesting question-answering  resource about COVID-19**

**Rating:** 6
**Confidence:** 5

**Review:**

This paper introduces a very interesting and useful question-answering resource about COVID-19. The corpus consists of 2200 frequent question-answer pairs, extracted from 40 trusted online sources plus 24k social media questions semantically-aligned (by experts) with one of the former dataset.

I really like the resource and think it is a good contribution to the workshop. Just have one suggestion regarding the manuscript and one advice about the chatbot that the authors intend to develop based on the corpus:

1. I missed some information on how BM25 was trained. Did you do it taking only the questions into account or you also use the answer linked to the candidate question?
2. Regarding the chatbot to be developed, it is important to notice that the answer to many questions  about COVID may change along the time. For instance, during quarantine, the question “Can I go to a bar?“ would be negative, whereas after the end of the lockdown would be positive. I am not sure how you can control the update of 2200 questions automatically. It is important to think about that in order to do not spread misinformation.

---

> ### Author Response · Authors · 2020-09-27
> **Response to AnonReviewer2**
>
> Thank you very much for your questions. We are happy to hear that you like the resource and think that it is a good contribution to the workshop. We are very excited about the opportunity for our work to be included in this workshop!
>
> 1. We trained the BM25 model by using the answers that were manually aligned by experts with the candidate questions. We then calculated scores between the terms in the input question and terms in the candidate answers. We used the implementation in Elasticsearch and relied on the default parameters.
>
> 2. We agree that not spreading misinformation (or old information that might have been thought to be true at one point but is no longer true) is very important. When developing our chatbot and designing our methods for collecting this dataset, this was a question that was often on our minds and hotly debated internally. In an earlier version of our schema (https://github.com/JHU-COVID-QA/scraping-qas/wiki/Schema-v0.1), we explicitly recorded the date that the question was first stored in our dataset, as well as when the data for when the question was last updated with a new answer.  To simplify our scraping process, we currently store the date a question-answer pair was added to our dataset and as well as the date for when we updated the answer to a question (dateLastChanged in https://github.com/JHU-COVID-QA/scraping-qas/wiki/Schema-v0.3).